# Monitoring of Spirulina Flakes and Powders from Italian Companies

**DOI:** 10.3390/molecules27103155

**Published:** 2022-05-14

**Authors:** Vanessa Dalla Costa, Raffaella Filippini, Morena Zusso, Rosy Caniato, Anna Piovan

**Affiliations:** Department of Pharmaceutical and Pharmacological Sciences, University of Padova, Via Marzolo 5, 35131 Padova, Italy; vanessa.dallacosta@phd.unipd.it (V.D.C.); raffaella.filippini@unipd.it (R.F.); morena.zusso@unipd.it (M.Z.); rosy.caniato@unipd.it (R.C.)

**Keywords:** *Arthrospira platensis*, microalgae, commercial products, morpho-chemical analysis, carotenoids, chlorophylls, pheophytins, phycobiliproteins, phenols, proteins

## Abstract

Microalgae and microalgae-derived compounds have great potential as supplements in the human diet and as a source of bioactive products with health benefits. Spirulina (*Arthrospira platensis* (Nordstedt) Gomont, or *Spirulina platensis*) belongs to the class of cyanobacteria and has been studied for its numerous health benefits, which include anti-inflammatory properties, among others. This work was aimed at comparing some spirulina products available on the Italian market. The commercial products here analyzed consisted of spirulina cultivated and processed with different approaches. Single-component spirulina products in powder and flake form, free of any type of excipient produced from four different companies operating in the sector, have been analyzed. The macro- and micromorphological examination, and the content of pigments, phycobiliproteins, phenols, and proteins have shown differences regarding the morphology and chemical composition, especially for those classes of particularly unstable compounds such as chlorophylls and carotenoids, suggesting a great influence of both culture conditions and processing methods.

## 1. Introduction

In recent years, microalgae are attracting high interest for their nutritional and therapeutic applications, being important sources of food ingredients and bioactive products with health benefits. Because of this, it is not difficult to imagine a substantial investment by food companies constantly looking for innovative products. This global trend clearly indicates the need for monitoring the quality of microalgal products, which are often promoted as “superfoods” although this term, used as a marketing tool, has no official definition by regulatory authority [1,2].

Among the microalgae, spirulina (*Arthrospira platensis* (Nordstedt) Gomont, or *Spirulina platensis*) is a filamentous cyanobacteria recognizable by the main morphological feature of the genus: the arrangement of the multicellular cylindrical trichomes in an open left-hand helix along the entire length [3]. Spirulina is the trade name of *Cyanobacteria* belonging to the genus *Arthrospira*, and food and food supplements are made from *A. platensis* and *A. maxima*. It is important to clarify that *Arthrospira* and *Spirulina* are two different genera, very similar from a morphological point of view but phylogenetically distant from each other [4]. The incorrect use of the trade name spirulina has always been confusing and the cause of frequent misunderstandings, even in the scientific field; nevertheless, this name is still commonly used.

Many of the studies on spirulina attempted to estimate its growth yield and photosynthetic efficiency, also in relation to its morphology. In fact, although the helical shape of the trichome is considered a stable and constant property maintained in culture, there may be considerable variation in the degree of helicity between different strains of spirulina and within the same strain. It has been observed that the ultrastructure may change from the typical helical shape to straight strain; the trigger and mechanism for this morphology switch is still unknown, but it was also proposed that the cell morphology was affected by environmental conditions such as nutrient availability, light, temperature, and salinity [3,5,6].

The interest in spirulina is linked to its very high nutritional value, so much that for the World Health Organization, spirulina is one of the greatest foods on earth [7]. The nutritional value is mainly due to its protein content (about 60%); spirulina also contains vitamins, minerals (iron, calcium, magnesium, zinc, manganese, phosphorus, and potassium), essential fatty acids (e.g., γ-linolenic acid, palmitic acid, linoleic acid, oleic acid), polysaccharides, glycolipids and sulfolipids, and enzymes (e.g., superoxide dismutase) [8,9]. 

Recently, spirulina has been widely studied for its numerous health benefits, which include antibacterial, antiviral, antioxidant, and anti-inflammatory properties [8,9,10,11,12,13,14]. Furthermore, many studies have evidenced the neuroprotective properties of spirulina in multiple models of CNS diseases, such as Parkinson’s disease, schizophrenia, ischemic brain damage and in lipopolysaccharide-induced neuroinflammation [15,16,17,18,19]. 

Most of the health benefits of spirulina are linked to the content of various bioactive pigments, including carotenoids, chlorophylls, pheophytins and phycobiliproteins [20,21,22]. 

Carotenoids play a variety of important functions and have been considered compounds able to fight the free radicals, reduce the risk of cancer, and prevent cardiovascular and neurodegenerative diseases, among others. The multiple health benefits of carotenoids are supposed to be due to their antioxidant and anti-inflammatory functions, considering that carotenoids can sequestrate free radicals released in the human body under stress conditions as well as reduce inflammatory mediators [23]. 

Phycobiliproteins are the key photosynthetic pigments found in various groups of cyanobacteria and red algae that capture light energy while protecting microalgae from harmful radiation. Phycobiliproteins include c-phycocyanin (blue pigment), allophycocyanin (light-blue pigment) and phycoerythrin (red pigment), which differ in their spectral properties [24]. The phycobiliproteins represent about 20% of the total protein content of spirulina. They are studied in recent years not only from a structural point of view but also with regard to their potential pharmacological activities. Indeed, it is reported that they have several health-promoting properties, including antioxidant, antibacterial, anticancer, anti-inflammatory, and immune-modulatory activities [25]. C-phycocyanin is also considered as a novel hypocholesterolemic protein [26].

Moreover, spirulina contains high levels of phenolic compounds, which contribute to the antioxidant activity of spirulina-based products [27,28,29].

The microalgal chemical composition depends on many factors, including strains, geographical origin, environmental conditions, and cultivation technologies. Moreover, other operational parameters were evaluated to relate them with spirulina product properties [30,31]. For example, researchers demonstrated a different functional value of spirulina biomass, depending on the processing methods used for its storage [32,33,34]. Park et al. [35] studied the major carotenoid and c-phycocyanin contents in commercially available spirulina powder products and laboratory-prepared spirulina trichomes. Results showed substantial differences in carotenoid and c-phycocyanin content between the commercially and laboratory-prepared samples. Hynstova et al. [36] quantified carotenoids and chlorophylls in dietary supplements samples in the form of powder and in the form of tablets. Differences in contents not related to powder or tablet form have been highlighted. 

Considering the variables, which may affect spirulina product characteristics, the assessment of the morpho-chemical features is the starting point for guaranteeing the production of safe ingredients and food supplements of high quality. 

In this study, *A. platensis* single-component products available on the Italian market, free of excipients, were analyzed. Four different spirulina-making brands of both flakes and powder of spirulina were selected to compare flake and powder-based preparations produced by the same company and to detect differences due to the cultivation method or the processing of the raw material. A macro- and micromorphological examination, and the analysis of the content of pigments, phenols, phycobiliproteins and proteins have been carried out. 

## 2. Results and Discussion

### 2.1. Commercial Product Selection

The selection of the studied spirulina products has been made evaluating different companies with the established Italian market position in the field of organic spirulina production. Particular attention was given to the cultivation, drying and processing techniques used to obtain the finished products, in order to assess the possible influence of these factors on the product characteristics. 

The information reported has been acquired through research on the websites of the companies and/or illustrative material and video provided by the companies themselves. Four companies have been selected: company 1 uses photobioreactors of the last generation, the biomass is separated by a membrane filtration system and dried at a temperature below 30 °C; company 2 uses indoor ponds, maintained at 60 °C, and the biomass is sifted and dried below 40 °C; company 3 uses indoor ponds maintained at 30 °C, and the biomass is separated and dried in thin layer below 40 °C; company 4 uses highly innovative systems, the algal growth is performed in raceways, the biomass filtration and drying is carried out in cleanrooms below 45 °C. 

The four selected companies differ in size and in chain of production, but all of them put on the market two product types: namely, single-component products based on spirulina in powder or flake form, free of excipients, that have been analyzed in this study. 

### 2.2. Macro-Microscopical and Olfactory Evaluation 

The macro- (A) and stereomicroscope analysis (B) of the samples underlined some differences (Figure 1). The flakes 1F were not made of flakes but of fragments with a cylindrical shape and green-bluish color; 2F and 3F had a similar shape, being made up basically of large thin flakes, but showed different color, bluish-green and brilliant green in 2F and 3F, respectively; 4F was much more fragmented, with a bright green color. The stereomicroscope observation showed that air pockets were present in the middle portion of the flake, more or less neatly distributed. The powders showed a high variability among the samples: 1P and 3P were constituted by a heterogeneous fine powder, 2P by an extremely fine powder, and 4P by a coarse powder. 

Algae have a great potential as new foods, but improvements need to be in place to enhance growth yields, nutritional quality, and organoleptic traits. A key to social acceptance may lie in the appeal of algae in terms of organoleptic traits, which may be essential to convincing people to consume algae products [37]. As regards olfactory characteristics, 1F has a light and sweet smell; it is similar but sharper in 1P. 2F and 2P have a characteristic algal smell, with a light fecal note. 3F and 3P had an algal smell with a mush note, which is more intense in 3P. 4F and 4P had a medium-intensity odor, lightly pungent and virose, which is more intense in 4P. 

*A. platensis* is recognizable by the arrangement of the multicellular cylindrical trichomes in an open left-hand helix along the entire length. In addition, straight or nearly straight variants were observed. The helical shape allows the trichome a better absorption of nutrients during movement in aquatic environments, whereas the straight structure improves the ability to move, increasing the speed of fluctuation and thus allowing it to migrate to more advantageous conditions if the environment is unfavorable. When, in a culture of a helically coiled strain, a few filaments happen to become straight, they tend to become predominant [38,39]. Environmental factors, mainly temperature, physical and chemical conditions, may affect the trichome geometry, and several studies consider straight trichomes to be a symptomatic feature of crops grown under stressful conditions, which are associated with slowed metabolism [40].

The analyzed samples differ between them for the trichome morphology (Figure 1C). As mentioned above, spirulina trichomes can have a helical to straight shape. The straight shape was prevalent in most of the samples analyzed. In 1F and 1P samples, the trichomes were only straight and very long; in 3F and 3P, they were mainly straight (90% ca.), and in 3P, they were even quite fragmented, which is most likely due to any damage that has occurred during the powdering process. In 2F and 2P samples, the straight and helical shapes coexisted almost equally (60% and 40%, respectively). Sample 4F showed straight trichomes only, while the corresponding powdered sample, 4P, appeared to be made up of trichomes almost exclusively in the helical shape. This could be due to a different harvest time; in fact, some authors reported that the straight filaments are peculiar in spirulina, which needs to be harvested [30]. 

### 2.3. Chemical Analyses

The chemical analyses were focused on those chemicals that, especially in recent years, have been the object of study by the scientific community for their increased interest in food and products with health benefits. The conditions relating to the extraction and analytical processes were chosen on the basis of preliminary experiments and literature data. Acetone extracts were made for carotenoid, chlorophyll and phaeophytin analysis, water extracts were made for phycobiliprotein analysis, and hydromethanol extracts were made for phenolic compound analysis. The total protein content was determined via elemental analysis.

#### 2.3.1. Carotenoids, Chlorophylls and Pheophytins

The acetone extracts showed very different color: 2P sample showed a very dark green color; 1F, 1P, 3F and 4F samples were bright green, and the other samples were of a more or less intense yellow hue (Figure 2).

HPLC UV-Vis analysis of the acetone extracts led to the identification of β-carotene, zeaxanthin and mixol 2′-methylpentoside; unidentified xanthophylls were also detected (Figure 3).

Figure 4 shows the relative content expressed as percentage. In all samples, β-carotene was the highest, which was followed by zeaxanthin, mixol 2′-methylpentoside and unidentified xanthophylls. Only in 3P, mixol 2′-methylpentoside was just above zeaxanthin.

Whereas the relative content profile was very similar and reflected the literature data, the content of total carotenoids was highly variable (Table 1).

The highest carotenoid content was 1.63 mg/g dw (sample 2P), and the lowest one was 0.24 mg/g dw (sample 1F). Except for samples 3F and 3P, powdered samples were more carotenoid-rich than the corresponding flakes. Samples 1F and 1P were the lowest in content. These results suggest that the differences in the total carotenoid content may be due to spirulina strain, growing conditions or processing techniques. However, the specific factors affecting the carotenoid content are not easy to be detected. It is also known that carotenoids show poor stability and are highly sensitive to oxidation and degradation during processing and storage, which further contribute to the high variability of their composition [41].

Figure 5 shows the content of chlorophylls and pheophytins. The data clearly indicate that 2P acetone extract, having a deep green color, has the highest content of chlorophylls and pheophytins (7.36 and 1.41 mg/g, respectively).

Pheophytins are a degradation product of chlorophylls, resulting from a process referred to as pheophytinization, which takes place during drying fresh biomass process by treatment with high temperature. Therefore, the ratio of chlorophylls and pheophytins can be used to evaluate the effects of raw material processing [36]. Table 2 shows the ratio of chlorophylls and pheophytins: no differences were found between the powdered and flaked samples produced by the same company. The values are very similar, only in samples F and P produced by brand 4 was the ratio lower. These data indicate that pheophytinization is less evident in brand 4, suggesting that the centralized management leading to the control and standardization of whole process, implemented by this company, may have reduced degradative reactions. However, these differences are minor, so the data suggest that the processing conditions used by the different companies did not substantially affect the breakdown of chlorophylls in pheophytins.

A fundamental role in the physiological processes of photosynthetic organisms has always been ascribed to chlorophylls and pheophytins. However, in more recent years, their potential health and pharmacological activities have been investigated, and it has been shown that they possess antioxidant activity, which has been demonstrated in different experimental models [42]. Their quantitative analysis has therefore become important in the evaluation of products with a potential application in the health/pharmaceutical field.

#### 2.3.2. Phycobiliproteins

The aqueous extracts had the characteristic light blue-blue color of the phycobiliproteins, and the color intensity was very different between the samples. Among the samples, 3P stood out with the lightest blue, and 1P and 2F had a medium intense blue (Figure 6).

Figure 7 shows the relative content of phycobiliproteins. In all the samples, c-phycocyanin was the most abundant component, which was followed by allophycocianin and phycoerythrin. The highest c-phycocyanin content was 101.5 mg/g dw (sample 2P), while the lowest one was 8 mg/g dw (sample 3P). A correlation between the type of product (flakes or powder) and c-phycocyanin content was not found. In fact, it should be noted that in samples 1 and 3, the powder had a lower content than the flakes, unlike samples 2 and 4, where the flakes were richer in c phycocyanin than powders.

It is known that phycobiliproteins and especially the most abundant c-phycocyanin serve as a nitrogen source during nitrogen starvation. Indeed, under favorable conditions for growth, c-phycocyanin is present in a large amount, whereas if the supply of nitrogen is low, about 30–50% of c-phycocyanin disappears without any effect on the maximal growth rate [43]. Therefore, it is feasible that the differences were due to different growth conditions or harvesting time.

The presence of red phycoerythrin was confirmed in all the samples, although its presence in spirulina is the subject of debate. While some studies report that small amounts of phycoerythrin are produced, others do not [44]. In all the samples analyzed in this study, phycoerythrin has been detected in an amount ranging from 1.21 to 9.5 mg/g dw.

#### 2.3.3. Phenolic Compounds

The hydromethanolic extracts showed an intense brown color in all the samples except for 1F and 1P, which were greenish (Figure 8).

The content of total phenolics was highly variable (Table 3). 

The highest phenolic content was 3.8 mg/g dw (sample 4F) and the lowest one was 1.27 mg/g dw (sample 1F). Except for sample 3, which shows a difference between flakes (2.26 mg/g dw) and powder (1.27 mg/g dw), there were no relevant differences between flakes and powder supplied by the same company.

The amount of phenolics produced by spirulina can be easily affected by environment factors, including the light irradiance intensity and culture medium [45,46]; moreover, phenolic compounds are very temperature-sensitive bioactives. According to Ouaabou [47], the main cause of a decrease in total phenolic content is the intense enzymatic activity of polyphenol oxidase (PPO) during drying process occurring up to 60 °C, while above this temperature, PPO is practically inactivated. All these interacting factors could therefore explain the different phenol contents found and reported in the literature. 

#### 2.3.4. Proteins

The total protein content in the analyzed samples is reported in Table 4.

The data obtained from the elemental analysis indicated percentages of N ranging from 8.70% to 9.69%. By the use of conversion factors, proteins are estimated without a demanding previous extraction, and possible losses of proteins are avoided during the preparation of the samples; in fact, it is known that differences in protein extraction procedures could have a remarkable influence on the final results [48,49,50,51]. Applying the nitrogen-to-protein conversion factors of 6.25 [52], which are commonly used to determine crude protein content in various matrices, protein values range from 53.4% (in 1F) to 60.6% (in 1P and 4F). These values are consistent with the average protein content (approximately 60%) reported in the literature. 

In spite of the conversion factor 6.25 being susceptible of error, as a result of non-proteinaceous nitrogen variability in different biomasses, it is still more applied for the majority of studies [53]. Because the aim of this study was the comparison of different samples and then the inter-product variability, it was considered that the results obtained could be sufficiently suitable to the purposed goal.

#### 2.3.5. Discrimination of the Samples

Table 5 summarizes the results obtained from the analyses.

The first focus is on the shape of the trichomes, which in the different samples is in form or exclusively straight, or both in the straight and helical form. It should be noted that no sample is exclusively made up of helical trichomes. In samples 1–3, the percentage of presence of the two forms of trichomes appears to be unchanged in the flakes and in the powder, unlike sample 4, where the flakes are constituted only by straight trichomes, while the powder almost exclusively by helical trichomes. As previously discussed, several authors showed that the straight form seems to be associated with some form of stress, which generally affects the level of metabolism. For example, it has been reported that an increased production of phenolic compounds is usually associated with biotic or abiotic insults. By observing the results obtained from chemical analysis, it can be noted that 1F and 1P samples, consisting only of straight trichomes, are among the poorest in metabolites, including the class of phenolic compounds. In contrast, sample 4F, which also consists only of straight trichomes, has one of the highest metabolite contents found in this study, and in this case, it has the highest phenolic compound content among all samples analyzed. The sample that by far has the highest content in the different classes of compounds, phenols excluded, is sample 2P, consisting of both straight and helical trichomes. Flakes, however, with similar morphology, are less rich than powder.

In order to discriminate the samples with respect to different companies and also different company products (flakes and powders), the data were subjected to principal component analysis. Figure 9 shows the corresponding score plots.

Two principal components were extracted explaining up to 91.39% of the total variance. At a glance, separation between the different samples can be observed. A clustering of the scores of samples 2F, 4F, and 4P occurs along F1 and F2 axes at positive values in the right upper side, while 1F and 1P scores clustered along the F1 and F2 axes at a negative value, in the left lower side. Sample 3P is at negative values of F1 and positive values of F2, while 3F and 2P are in the right lower side, at positive values of F1 and negative values of F2.

Only samples from companies 1 and 4 powders and flakes appear in the same square.

## 3. Materials and Methods

### 3.1. Materials

Eight commercially available spirulina products from four different Italian companies (1, 2, 3, 4) were analyzed. For each company, two products were taken into account: flakes (S) and powder (P), both of which consisted only of dried seaweed grown by the company.

### 3.2. Macro- and Microscopic Characterization 

The first observations focused on dried samples were made directly with the naked eye and by a stereomicroscope S9i (Leica). The samples used for optical microscope observation were prepared by transferring equal aliquots of each sample to an Eppendorf with a known volume of deionized water. The samples were gently shaken, and the cell suspension was placed on a microscope slide. Microscopic images were acquired by an inverted light microscope Telaval 31 (Carl Zeiss), and the images were acquired at 400× magnification (10× eyepiece, 40× objective). 

### 3.3. Preparation of Spirulina Extracts

For sample preparation, the original products were grinded using a mortar and pestle, and the obtained powder was extracted as described below. 

*Acetone extracts*: First, 0.5 g of each sample was extracted using 25 mL acetone in an ultrasound bath for 30 min in an ice bath. Extraction was carried out in the absence of light. The samples were placed in the fridge at 4 °C for 24 h [30, modified] and after centrifugation at 13.2 rpm for 4 min at 4 °C, the supernatants were analyzed. 

*Aqueous extracts*: First, 1 g of each sample was extracted with 25 mL of milliQ water in an ultrasound bath for 40 min [25]. The samples were then centrifuged at 13.2 rpm for 10 min at 20 °C, and the supernatants were analyzed.

*Hydromethanolic extracts*: First, 0.5 g of each sample was extracted with 10 mL of 80% hydromethanolic solution by stirring (104 rpm) at 25 °C for 24 h. The samples were centrifuged at 13.2 rpm for 7 min at 20 °C, and the supernatants were analyzed [30, modified].

### 3.4. Analysis of Carotenoids 

An aliquot of acetone extract was dried under vacuum and solubilized in methanol. The methanol solutions were analyzed using an Agilent 1100 HPLC Series System (Agilent, Santa Clara, CA, USA) equipped with a degasser, quaternary gradient pump, column thermostat, and UV-Vis detector. A Gemini 5 µm C6-Phenyl column (250 × 4.6 mm) from Phenomenex (Torrance, CA, USA) was set at 40 °C. Analyses were conducted in the isocratic mode, using acetonitrile/methanol (10:90; *v*/*v*) at a flow rate of 1 mL min−1, with an injection volume of 10 µL; detection was performed at 460 nm. Spectra were acquired from 190 to 800 nm. The identification of the compounds was performed by comparing HPLC chromatograms and UV-Vis spectra using the available reference standards (zeaxanthin, β-carotene: Sigma-Aldrich, Milan, Italy) and data obtained from published references [36,54].

Since standards were not available for all carotenoids, zeaxanthin and β-carotene were selected as external standards for the quantification. Stock solutions (1 mg/mL) were prepared in acetonitrile/methanol (10:90; *v*/*v*), and the calibration curves were obtained in a concentration range, respectively, of 0.25–25 μg/mL (R^2^ = 0.9996), 5–30 μg/mL (R^2^ = 0.9998), with six concentration levels. Calibration curves were constructed by plotting the peak area at 460 nm vs. the pigment concentrations. Xanthophylls were quantified as zeaxanthin equivalents. All the samples were analyzed in triplicate and analyzed twice a week apart; the results were reported as means ± standard deviation (SD). The amount of the compounds was expressed as mg/g of the dry weight of spirulina.

### 3.5. Analysis of Chlorophylls and Pheophytins

Chlorophyll a and chlorophyll b content was determined as previously described [23]. Pheophytin total content was measured according to the method of Hynstova et al. [36]. An aliquot of the acetone extracts was appropriately diluted with acetone:water (4:1), and the maximum absorbance was read at 663, 646 nm for chlorophyll a and chlorophyll b, and 653, 654 nm for pheophytins using a HeλIOS α (Thermo Electron Corporation) spectrophotometer. The content of pigments was calculated using the following equations:Chlorophyll a (µg/mL) = 12.25 A_663_ − 2.25 A_646_
Chlorophyll b (µg/mL) = 20.31 A_646_ − 2.25 A_663_
Total pheophytins (µg/mL) = 321,3 A_653_ − 208,4 A_654_


All the samples were analyzed in triplicate and analyzed twice a week apart; the results were reported as means ± standard deviation (SD). The amount of the compounds was expressed as mg/g of the dry weight of spirulina.

### 3.6. Analysis of Phycobiliproteins

C-phycocyanin, allophycocyanin and phycoerythrin content was determined as described by Aouir et al. [30]. One aliquot of the aqueous extracts was appropriately diluted, and the maximum absorbance was read at 620, 652 nm for c-phycocianin and allophycocianin, 562 nm for phycoerythrin, using a HeλIOS α (Thermo Electron Corporation) spectrophotometer. The content of phycobiliproteins was calculated using the following equations:c-phycocyanin = [A_620_ − (0.474 · A_652_)]/5.34
allophycocyanin = [A_652_ − (0.208 · A_620_)]/5.09
phycoerythrin = [A_562_ − (2.41 c-phycocyanin)] − (0.849 allophycocyanin)]/9.62

All the samples were analyzed in triplicate and analyzed twice a week apart; the results were reported as means ± standard deviation (SD). The amount of the compounds was expressed as mg/g of the dry weight of spirulina.

### 3.7. Analysis of Phenols 

Total phenolic content was determined as described by Li et al. [55]. First, 100 µL microliters of hydromethanolic appropriately diluted extracts were added to 1 mL of 1:10 diluted Folin–Ciocalteu reagent. After 4 min, 800 μL of saturated sodium carbonate (75 g/L) was added. After 2 h of incubation at room temperature, the absorbance at 765 nm was measured using a HeλIOS α (Thermo Electron Corporation) spectrophotometer. Gallic acid (2–10 µg/mL) was used for the standard calibration curve (R^2^ = 0.9999). The results were expressed as gallic acid equivalent (GAE) mg/g dry weight of spirulina. All the samples were analyzed in triplicate and analyzed twice a week apart; the results were reported as means ± standard deviation (SD).

### 3.8. Protein Content

The total protein content was determined on the grinded powder by a vario microCube (Elementar) elemental analyzer; the values obtained were multiplied by the conversion factor 6.25 [56] and expressed as g/100 g dry weight of spirulina.

### 3.9. Statistical Analysis 

Principal component analysis was carried out using XLSTAT.

## 4. Conclusions

Experimental studies and reviews on spirulina are very numerous, and often, the comparison of the reported data is difficult. As far as chemical composition is concerned, the attention of the scientific community initially focused on protein content and, in more recent years, also on pigment content and, to a lesser extent, on phenolic compounds.

The first difficulty is the diversity in analytical methods used, which often precludes a meaningful comparison. The second difficulty is linked to the starting material; the existence of different strains, cultural conditions, and processing methods often makes an end in itself the qualitative–quantitative determination of a product based on spirulina. However, if you want to support a rational approach to food supplements and/or herbal therapies, we need to understand how we can obtain standardized products, ensuring the consumer consistency of composition, which is an essential requirement for consistency of effectiveness and, even more importantly, safety.

In this work, eight spirulina products were analyzed in the form of flakes and powder, free of excipients, purchased from four companies, to assess their morphological characteristics and chemical profile, in order to understand if any differences could be rationally ascribed to the different production methods, which are available in detail on the websites of the companies themselves.

On the basis of the results obtained, it is not possible to hypothesize the correlations between morphology, type of product (flakes and powder) and content in metabolites. The observed differences, even when considerable, suggest a great influence on the morphology and chemistry of the culture conditions. Processing methods can also have a major influence on chemical composition, especially for those classes of particularly unstable compounds such as chlorophylls and carotenoids. Moreover, the simple visual observation of the samples and their extracts foreshadowed a variability in content, being spirulina particularly rich in pigments; the 3P sample, the clearest in color, is also one of the less rich samples among the studied samples. 

Spirulina has a long tradition of use, a great importance from an economic point of view, and a great potential in food and pharmaceutical fields. It is therefore more necessary than ever to clarify some fundamental aspects related to its morpho-chemical characteristics, because these are the starting point to ensure the quality of the products, but also because the development of “high-tech” products is reduced to a claim not reflected in reality.

## Figures and Tables

**Figure 1 molecules-27-03155-f001:**
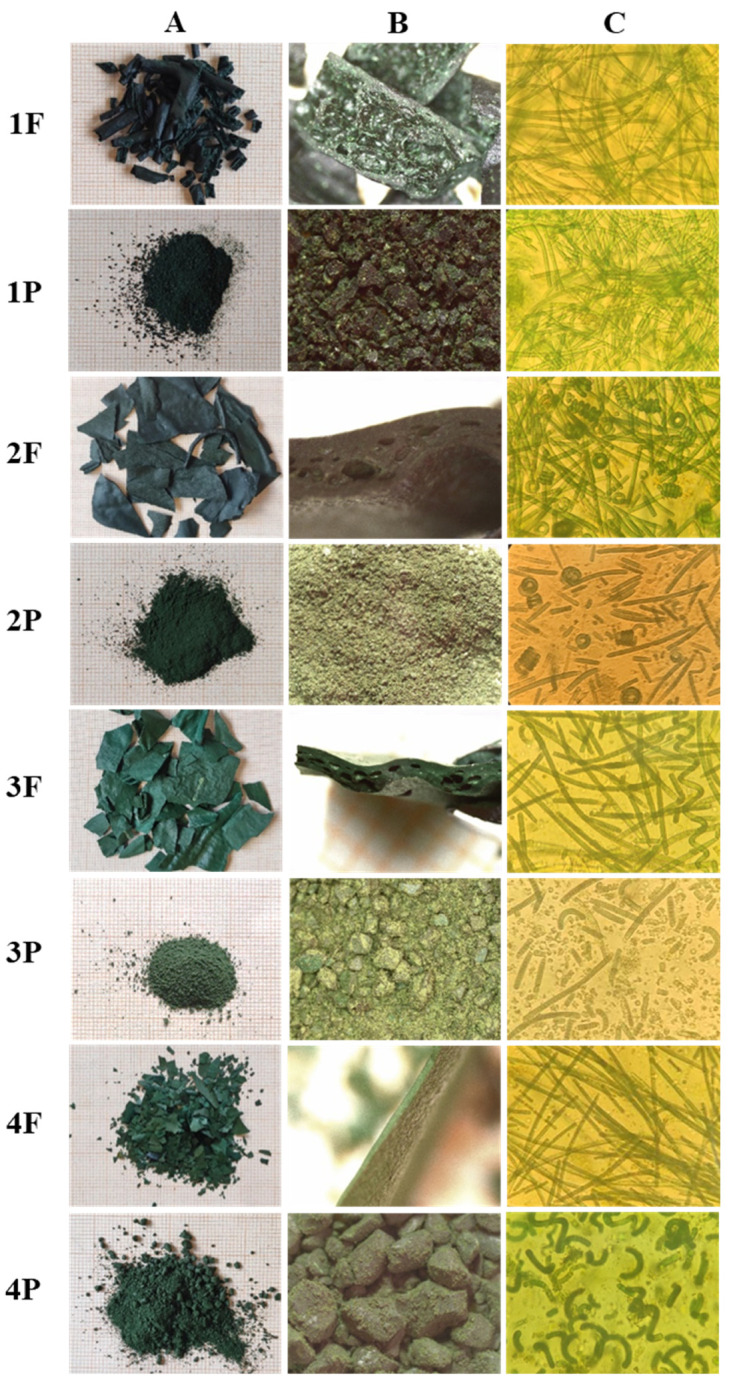
(**A**) Macroscopic images; (**B**) images acquired by stereomicroscope; (**C**) images acquired by optical microscope.

**Figure 2 molecules-27-03155-f002:**
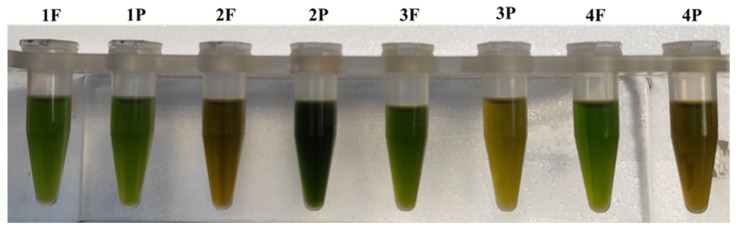
Acetone extracts.

**Figure 3 molecules-27-03155-f003:**
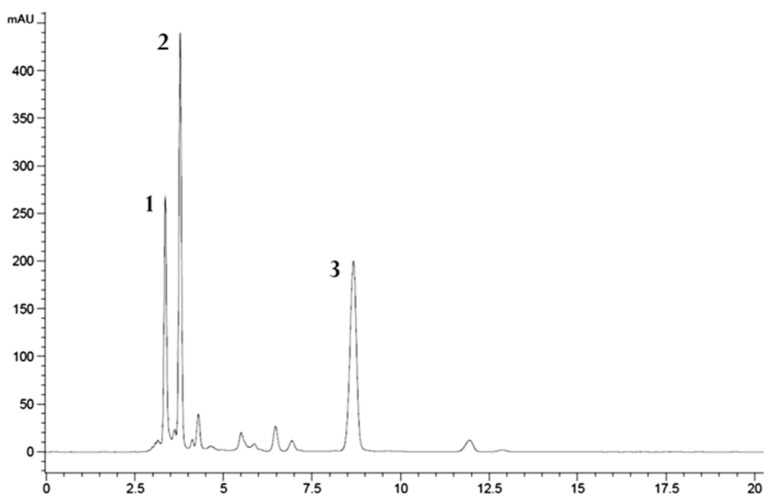
Chromatogram of 2P acetone extract acquired at 460 nm: mixol 2’-methylpentoside (1), zeaxanthin (2) and β-carotene (3).

**Figure 4 molecules-27-03155-f004:**
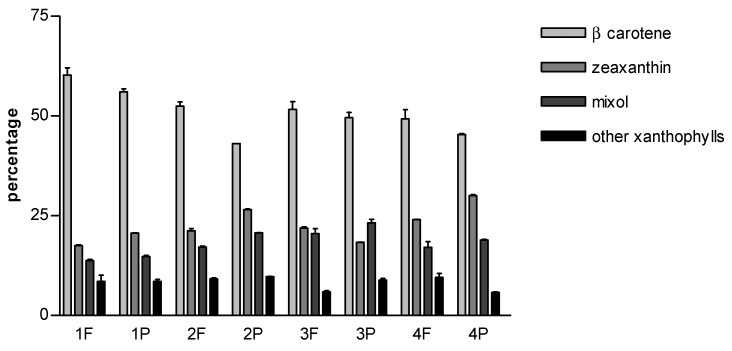
Relative carotenoid content expressed as percentage.

**Figure 5 molecules-27-03155-f005:**
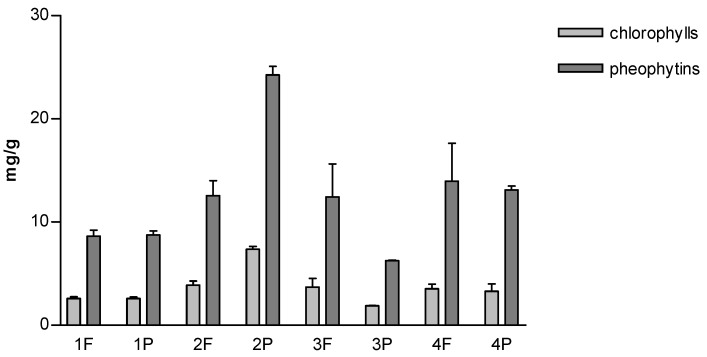
Content of chlorophylls and pheophytins expressed as mg/g dry weight of spirulina.

**Figure 6 molecules-27-03155-f006:**
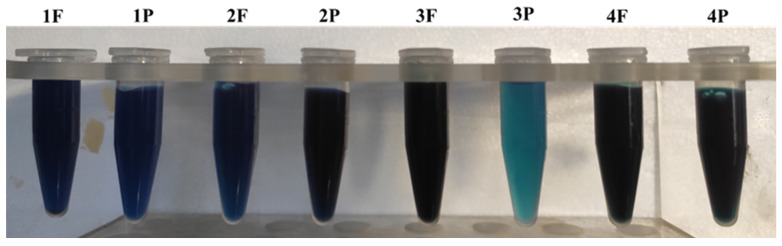
Aqueous extracts.

**Figure 7 molecules-27-03155-f007:**
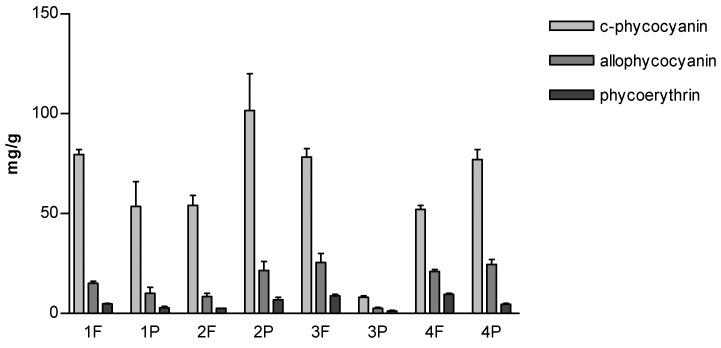
Content of phycobiliproteins expressed as mg/g dry weight of spirulina.

**Figure 8 molecules-27-03155-f008:**
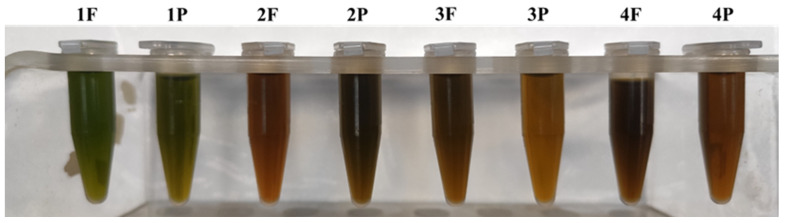
Hydromethanolic extracts.

**Figure 9 molecules-27-03155-f009:**
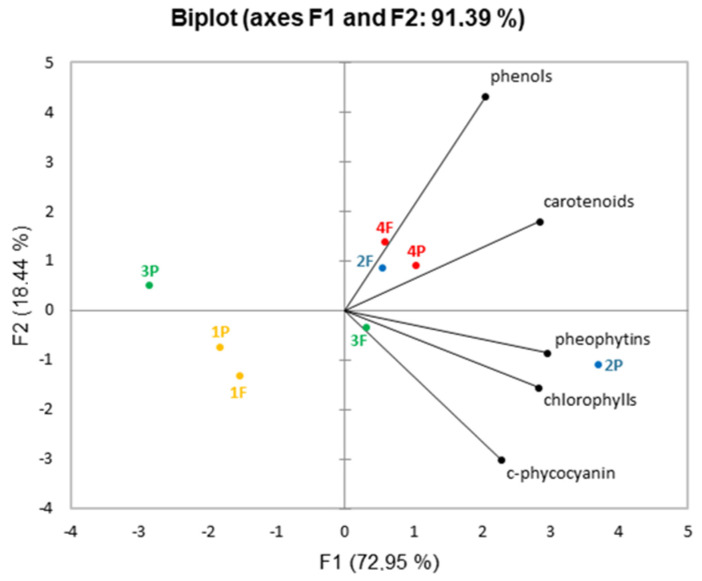
Biplot of principal component analysis.

**Table 1 molecules-27-03155-t001:** Total carotenoid content expressed as mg/g dry weight of spirulina.

*Samples*	1F	1P	2F	2P	3F	3P	4F	4P
**Carotenoids**	0.24 ± 0.01	0.32 ± 0.02	1.28 ± 0.18	1.63 ± 0.08	1.04 ± 0.35	0.37 ± 0.00	1.01 ± 0.30	1.41 ± 0.08

**Table 2 molecules-27-03155-t002:** Ratio of chlorophylls and pheophytins.

*Samples*	1F	1P	2F	2P	3F	3P	4F	4P
**Chlorophylls/Pheophytins**	0.30	0.30	0.31	0.30	0.30	0.30	0.25	0.25

**Table 3 molecules-27-03155-t003:** Total phenol content expressed as gallic acid equivalent (GAE) mg/g dry weight of spirulina.

*Samples*	1F	1P	2F	2P	3F	3P	4F	4P
**Phenols**	1.27 ± 0.06	1.30 ± 0.03	3.04 ± 0.01	2.67 ± 0.02	2.26 ± 0.16	1.47 ± 0.03	3.84 ± 0.11	3.48 ± 0.13

**Table 4 molecules-27-03155-t004:** Total protein content expressed as g/100 g dry weight of spirulina.

*Samples*	1F	1P	2F	2P	3F	3P	4F	4P
**Proteins**	54.38	60.56	59.94	60.06	54.94	58.00	60.56	56.13

**Table 5 molecules-27-03155-t005:** Comparation among trichome shape, carotenoids, chlorophylls, pheophytins, phenols and phycobiliproteins. Within the same class of compounds, value “1” has been assigned to the lowest value, and to the others, a multiplication factor n-fold higher as compared to the lowest value.

	TRICHOMES %Straight %Helical		CAROTENOIDS	CHLOROPHYLLS	PHEOPHYTINS	PHYCOBILIPROTEINS	PHENOLS
**1F**	100	0	**1**	1.4×	1.4×	8.4×	**1**
**1P**	100	0	1.4×	1.4×	1.4×	5.6×	**1**
**2F**	60	40	5.3×	2.0×	2.0×	5.5×	2.4×
**2P**	60	40	6.8×	3.9×	3.9×	11.0×	2.1×
**3F**	90	10	4.3×	1.9×	2.0×	9.6×	1.8×
**3P**	90f	10f	1.5×	**1**	**1**	**1**	1.2×
**4F**	100	0	4.2×	1.9×	2.2×	7.0×	3.0×
**4P**	10	90	5.9×	1.7×	2.0×	9.0×	2.7×

f = fragmented.

## Data Availability

Not applicable.

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
