# Peer review of "Monitoring of Spirulina Flakes and Powders from Italian Companies"

_molecules, 2022, doi:10.3390/molecules27103155_

Round 1

Reviewer 1 Report

The submitted work dealt with the chemical analysis of eight spirulina products from Italian market. Based on the obtain results, it is suggested that a great influence both of the culture conditions and the processing methods on the morphology and chemical composition. Although the motivation is of interest, the experimental design and quality of analytical experiments did not reach the Molecules journal standard. I would suggest to include a comprehensive analytical work on more metabolic variance and more commercial products. Therefore, I did not suggest to accept the manuscript in the current form. Few minor concerns are as follows:

Line 10: important

Line 28: repeated sentences with abstract “in recent years, microalgae are gaining a high interest for their nutritional and therapeutic applications being important sources of food ingredients and bioactive products with health benefits.”

Line 46: γ-linoleic acid is belonging to linoleic acid derivatives?

Line 49: please state “anti-” or “anti” consistently throughout the whole manuscript (eg. Anti-oxidant and anti-inflammatory or antioxidant and antiinflammatory)

Line 53: please cite the references to support this claim

Line 63: c-phycocyanin

Line 66: …In spirulina.

Line 78: conditions [24].

All tables and text: all values should be written as “0.24 ± 0.01” not “0,24±0,01”

Line 305: Tabella 5?

Author Response

Dear Reviewer,

we hope that the revisions will be satisfactory.

The submitted work dealt with the chemical analysis of eight spirulina products from Italian market. Based on the obtain results, it is suggested that a great influence both of the culture conditions and the processing methods on the morphology and chemical composition. Although the motivation is of interest, the experimental design and quality of analytical experiments did not reach the Molecules journal standard. I would suggest to include a comprehensive analytical work on more metabolic variance and more commercial products.

As suggested we have improved the experimental design adding a PCA analysis of the data (Figure 9).

So much for the number of samples analysed, as reported in the text, the selection has been made evaluating different companies with established Italian market position in the field of organic spirulina production. The choice was made by taking in consideration the companies producing crude flakes and powders excipient-free as human food, by using   different cultivation, drying and processing techniques. 

Therefore the goal of the work lies precisely in the choice of samples, only spirulina flakes and powders differently processed from Italian companies, which differs from the random choice of commercial products.

The metabolic variance is in line with other published works on Molecules, e.g. Park WS et al. Molecules 2018, 23, 2065; doi:10.3390/molecules23082065; Papalia et al., Molecules 2019, 24, 2810; doi:10.3390/molecules24152810.

We have revised the manuscript as suggested:

Line 10: important. Done

Line 28: repeated sentences with abstract “in recent years, microalgae are gaining a high interest for their nutritional and therapeutic applications being important sources of food ingredients and bioactive products with health benefits.” Replaced.

Line 46: γ-linoleic acid is belonging to linoleic acid derivatives? The reported γ-linoleic was a typing mistake.

Line 49: please state “anti-” or “anti” consistently throughout the whole manuscript (eg. Anti-oxidant and anti-inflammatory or antioxidant and antiinflammatory). Used “anti”.

Line 53: please cite the references to support this claim. Done (new Ref. 17-19).

Line 63: c-phycocyanin. Not understood.

Line 66: …In spirulina. Done.

Line 78: conditions [24]. Done.

All tables and text: all values should be written as “0.24 ± 0.01” not “0,24±0,01”. Done.

Line 305: Tabella 5? Corrected.

Reviewer 2 Report

The article is very interesting, but it does contain some inaccuracies. They concern:
title: should be a bit shorter, in its current form it is quite illegible.
This is also related to the suggestion for keywords, which unfortunately are a repeat of the title. In addition, they should be phrases, the most important keys by which the reader can easily find the article in search engines. However, they should not duplicate the title. The authors are asked to correct these inaccuracies.
My point is also about the lack of statistical analysis. In this type of research, it is important and allows for a better interpretation of the results.

Author Response

Dear Reviewer,

we hope that the revisions will be satisfactory.

The article is very interesting, but it does contain some inaccuracies. They concern:
title: should be a bit shorter, in its current form it is quite illegible.
This is also related to the suggestion for keywords, which unfortunately are a repeat of the title. In addition, they should be phrases, the most important keys by which the reader can easily find the article in search engines. However, they should not duplicate the title. The authors are asked to correct these inaccuracies. My point is also about the lack of statistical analysis. In this type of research, it is important and allows for a better interpretation of the results.

As suggested, we have revised the title and the key words, which we hope will be satisfactory.

Furthermore, we followed Your tip adding a PCA analysis of the data (Figure 9).

Reviewer 3 Report

The aim of a quality control of products containing microagae is relevant and necessary considering their importance in the market, but several aspects of the paper must improuved. In particular, several key published data and references were not considered. Some papers available in the literature present images of similar products and they should be carefully considered. Furthermore, because some of the reported images are not fully in accordance, considering the high presence of linear colonies. Unfortunatly, the resolution is not sufficient to exclude the presence of toxic microalgae. A deep revision of the paper is necessary, including a deep discussion of presented data and comparison with similar reserches. 

Author Response

Dear Reviewer,

we hope that these revisions will be satisfactory.

The aim of a quality control of products containing microalgae is relevant and necessary considering their importance in the market, but several aspects of the paper must improved. In particular, several key published data and references were not considered. Some papers available in the literature present images of similar products and they should be carefully considered. Furthermore, because some of the reported images are not fully in accordance, considering the high presence of linear colonies. Unfortunatly, the resolution is not sufficient to exclude the presence of toxic microalgae. A deep revision of the paper is necessary, including a deep discussion of presented data and comparison with similar researches. 

As suggested, we have taken in consideration some further papers and improved the introduction (new Ref. 17-19, 28). As we have observed on the analyzed samples, is well reported the presence of both helical and straight forms of Arthrospira. The shift from helical to linear form is not entirely clear, but once a strain converts to linear form it fails to return helical, unless with very specific growth conditions, and this reversion seems rather rare and happens only for some strains.

In addition, the presence of toxic microalgae can be ruled out because the companies are authorized to produce spirulina as food for the citizen consumption. In Italy, as in the European Union, all the food production is under the EU regulation n.178/2002, in which the principle of food safety, from the start to the end of supply chain, is followed.

Round 2

Reviewer 3 Report

Previous comments were only partially satisfied. The comparison with other papers, practically focused on the same subject should be carefully considered by the authors.

Author Response

Dear Reviewer,

we wish to thank You for Your suggestions. We are now submitting the revised version of the manuscript.

Your comment: “Previous comments were only partially satisfied. The comparison with other papers, practically focused on the same subject should be carefully considered by the authors”.

We have reorganized and improved the introduction, taking in consideration other papers focused on the same subject (see ref. 1-3, 5, 32-35).

Yours sincerely,

Anna Piovan